# Drug Exposure and Susceptibility of Pyrazinamide Correlate with Treatment Response in Pyrazinamide-Susceptible Patients with Multidrug-Resistant Tuberculosis

**DOI:** 10.3390/pharmaceutics16010144

**Published:** 2024-01-21

**Authors:** Shulan Dong, Ge Shao, Lina Davies Forsman, Sainan Wang, Shanshan Wang, Jiayi Cao, Ziwei Bao, Judith Bruchfeld, Jan-Willem C. Alffenaar, Jia Liu, Yi Hu, Meiying Wu

**Affiliations:** 1Department of Epidemiology, School of Public Health and Key Laboratory of Public Health Safety, Fudan University, Shanghai 201203, China; 23211020056@m.fudan.edu.cn (S.D.); shaog21@m.fudan.edu.cn (G.S.); 21211020018@m.fudan.edu.cn (S.W.); 22211020184@m.fudan.edu.cn (S.W.); 22211020106@m.fudan.edu.cn (J.C.); 2Department of Medicine, Division of Infectious Diseases, Karolinska Institutet Solna, 171 77 Stockholm, Sweden; lina.davies.forsman@ki.se (L.D.F.); judith.bruchfeld@ki.se (J.B.); 3Department of Infectious Diseases, Karolinska University Hospital, 171 76 Stockholm, Sweden; 4Department of Infectious Diseases, The Fifth People’s Hospital of Suzhou, Suzhou 215007, China; baoziwei521@126.com (Z.B.); wu_my@126.com (M.W.); 5School of Pharmacy, Faculty of Medicine and Health, University of Sydney, Sydney 2006, Australia; johannes.alffenaar@sydney.edu.au; 6Westmead Hospital, Sydney 2145, Australia; 7Sydney Infectious Diseases Institute, University of Sydney, Sydney 2006, Australia

**Keywords:** pyrazinamide, multidrug-resistant tuberculosis, pharmacokinetics, minimum inhibitory concentration, treatment outcome

## Abstract

Exploring the influence of pyrazinamide exposure and susceptibility on treatment response is crucial for optimizing the management of multidrug-resistant tuberculosis (MDR-TB). This study aimed to investigate the association between pyrazinamide exposure, susceptibility, and response to MDR-TB treatment, as well as find clinical thresholds for pyrazinamide. A prospective multi-center cohort study of participants with MDR-TB using pyrazinamide was conducted in three TB-designated hospitals in China. Univariate and multivariate analyses were applied to investigate the associations. Classification and Regression Tree (CART) analysis was used to identify clinical thresholds, which were further evaluated by multivariate analysis and receiver operating characteristic (ROC) curves. The study included 143 patients with MDR-TB. The exposure/susceptibility ratio of pyrazinamide was associated with two-month culture conversion (adjusted risk ratio (aRR), 1.1; 95% confidence interval (CI), 1.07–1.20), six-month culture conversion (aRR, 1.1; 95% CI, 1.06–1.16), treatment success (aRR, 1.07; 95% CI, 1.03–1.10), as well as culture conversion time (adjusted hazard ratio (aHR) 1.18; 95% CI,1.14–1.23). The threshold for optimal improvement in sputum culture results at the sixth month of treatment was determined to be a pyrazinamide AUC_0–24h_/MIC ratio of 7.8. In conclusion, the exposure/susceptibility ratio of pyrazinamide is associated with the treatment response of MDR-TB, which may change in different Group A drug-based regimens.

## 1. Introduction

Multidrug-resistant tuberculosis (MDR-TB), defined as tuberculosis (TB) resistant to rifampicin and isoniazid, is still a threat to the worldwide control of TB [1]. In the past few decades, significant progress has been made, such as more effective treatment regimens including new and repurposed anti-TB drugs [2]. In 2016, the WHO recommended an 18–24-month all-oral regimen including bedaquiline (BDQ), linezolid (LZD), and moxifloxacin (MFX) as Group A drugs, in which pyrazinamide was classified as a Group C agent [3]. Later, pyrazinamide was recommended in short-course all-oral regimens for MDR-TB in 2020 [4]. Although the treatment success rate keeps increasing, MDR-TB treatment still entails long treatment periods and limited drug options. Therefore, it is important to optimize the use of currently available drugs through drug exposure and pathogen susceptibility testing.

Pyrazinamide is often used as a potent first-line sterilizing agent against Mycobacterium tuberculosis (*M. tuberculosis*) both in China and worldwide. It has a unique efficacy against persistent and semi-dormant bacilli [5,6], which enables a shortened treatment length and a decrease in relapse rates. Both in long- and short-course regimens of MDR-TB treatment, pyrazinamide has demonstrated its potential to improve treatment outcomes in MDR-TB and reduce the time of sputum culture conversion [4,7,8] unless there is confirmed resistance.

Treatment responses in participants with MDR-TB can be partly explained in terms of drug exposure and *M. tuberculosis* susceptibility [9]. Pharmacokinetic parameters are important indicators for evaluating exposure levels. Among them, the 0–24 h area under the concentration–time curve (AUC_0–24h_) was used to estimate the total drug absorption [10]. Given the standard doses (20–30 mg/kg) of pyrazinamide, it typically falls within the range of 200 to 400 mg·h/L. It is influenced by factors such as sex, wight, liver function, comorbidities, and other drugs [11]. An adequate pyrazinamide AUC_0–24h_ is important for effective therapy, while a suboptimal level is correlated with unfavorable treatment outcomes [10,12,13]. Previous studies indicated that pyrazinamide susceptibility is associated with better treatment responses [6,8,14]. However, the proportion of pyrazinamide resistance ranges from 10% to 85% among patients with MDR-TB in different geographic populations [15,16,17,18], which indicates the necessity of considering how to better use pyrazinamide.

Therapeutic drug monitoring (TDM) is a tool for optimizing and individualizing dosage [19]. Previous studies widely used the ratio of AUC_0–24h_ to the minimum inhibitory concentration (MIC) as a clinical threshold in TDM during TB treatment [20]. In 2014, Chigutsa et al. [21] recommended pyrazinamide AUC_0–24h_/MIC at 11.3 based on the β-slope, which reflects sterilizing activity, in patients with drug-susceptible TB (DS-TB). In 2021, our research group [22] derived its threshold to be 8.42 by CART analysis in Chinese patients with DS-TB. Subsequently, a similar study [9] was conducted in Chinese patients with MDR-TB but did not derive a clinical threshold of pyrazinamide due to limited pharmacokinetic data. Considering the difference between DS-TB and MDR-TB regimens, the threshold of pyrazinamide may require re-evaluation with respect to the currently recommended regimen in MDR-TB treatment.

Based on the aforementioned situation, it is of great importance to determine the clinical target of pyrazinamide in patients with MDR-TB. To fill this knowledge gap, the objective of this study was to identify the correlation between pyrazinamide exposure, susceptibility, and treatment responses. We subsequently applied machine learning algorithms to identify the clinical thresholds of pyrazinamide to predict treatment efficacy in patients with MDR-TB using the current WHO-recommended long-term treatment regimen [4].

## 2. Materials and Methods

### 2.1. Study Design and Participants

We conducted a multicenter prospective cohort study from July 2019 to June 2020 in three hospitals in Henan, Jiangsu, and Guizhou Province in China as previously reported [9,22]. We selected the field sites that represented varying levels of socioeconomic development and TB burden. Jiangsu province was well economically developed with a low TB burden, while Guizhou and Henan were less developed with a relatively higher TB burden. The subjects of the present study were adult (aged ≥18 years) patients with MDR-TB. Considering the objective of this study, we included patients with *M. tuberculosis* isolates susceptible to pyrazinamide and used the treatment regimens including it. Patients were excluded if they refused to participate. To avoid the impacts of possible confounders, we excluded patients aged >70 years; pregnant; diagnosed with hepatitis B, C virus, or HIV; diagnosed with clinically significant abnormal renal or liver injury; and having been treated for MDR-TB over one day. This study included all patients who met the inclusion and exclusion criteria and provided written consent during the study period.

This study was approved by the Ethics Committee of the School of Public Health, Fudan University (Shanghai, China; 2018-06-0929), and written informed consent was obtained from all subjects.

### 2.2. MDR-TB Treatment and Information Collection

Upon diagnosis of MDR-TB, participants underwent inpatient treatment for two weeks, followed by outpatient treatment in designated hospitals. The standardized long-term oral regimen consists of a six-month intensive phase (mainly using fluoroquinolones, bedaquiline, linezolid, clofazimine, and cycloserine) and an 18-month continuation phase (mainly using fluoroquinolones, linezolid, clofazimine, and cycloserine). All the participants were treated with regimens based on susceptibility and availability of the anti-TB drugs according to the WHO and national guidelines. All regimens consisted of five effective drugs including pyrazinamide [4]. Due to the observational properties of this cohort study, no intervention was implemented and pharmacokinetic information was not used to adjust treatment.

We used a questionnaire to collect demographic characteristics, while hospital records were reviewed for clinical information as well as drug regimen and dosage. To record treatment interruption or missing doses as well as other reasons, nurses directly observed drug intake during the inpatient treatment. Community healthcare workers observed that in the outpatient therapy. During the 24 months of follow-up, examinations were conducted every month in the intensive phase and every two months in the continuation phase.

### 2.3. Drug Susceptibility Testing

We collected participants’ sputum samples at each visit and sent them to the reference laboratories with a well-controlled quality of TB analysis. Bacterial culture was conducted in the BACTEC MGIT 960 system (Becton Dickinson, Franklin Lakes, NJ, USA) [23,24]. Culture time to positivity (TTP) was considered a marker of *M. tuberculosis* load. The positive inoculum was further used for phenotypic drug susceptibility testing (pDST) in the same system. Pyrazinamide susceptibility (Sigma-Aldrich, Darmstadt, Germany) was determined with a pH of 5.9 considering the specific pH requirement for effectiveness. The critical concentration used for the classification of pyrazinamide drug susceptibility was 100 mg/L in accordance with the WHO technical guideline [25].

Then, the inocula of strains identified as pyrazinamide-susceptible via the pDST were further transferred onto Lowenstein–Jensen medium to determine the minimum inhibitory concentration (MIC) of pyrazinamide, the lowest concentration of pyrazinamide to inhibit *M. tuberculosis* growth. The concentration used for testing of pyrazinamide was 1–100 mg/L. Since only pyrazinamide-susceptible isolates were included, the range of MICs for statistical calculations was limited to 0–100 mg/L. Please refer to our previously published study [22] for more details on the experimental procedure.

### 2.4. Drug Exposure

We used venous catheters to collect participants’ blood samples initially at pre-dose as well as 1, 2, 4, 6, 8, 10, 12, 16, and 18 h after the observed intake of anti-TB drugs after two weeks of inpatient treatment [26]. Blood samples were sent for chromatographic separation and gradient elution on a Zorbax Aq-SB high-performance liquid chromatography (HPLC) column (internal diameter: 2.1 × 50 mm; particle diameter: 1.8 μm; Agilent Technologies, Santa Clara, CA, USA). The mobile phase was water containing 0.1% formic acid (50:50, *v*/*v*) and acetonitrile and the flow rate was set to 0.3 mL/min. Then, we used the liquid chromatography–tandem mass spectrometry method (LC–MS/MS) to measure the concentration of pyrazinamide based on the previously established and validated method [27] with pyrazinamide-d3 as the internal standard. The detection of pyrazinamide was based on the multiple reaction monitoring of *m*/*z* 124.1→81.0 operated in positive ionization mode. Since only phenotypically pyrazinamide-susceptible isolates were included, the range of MICs for analysis was limited to 0.1–100 mg/L. The inter- and intra-day variation was 5–15% with r^2^ >0.99. The pyrazinamide AUC_0–24h_ was calculated using the non-compartmental analysis tool of the Phoenix WinNonlin^®^ software (version 8.3, Pharsight Corporation, Saint Louis, MO, USA) with the trapezoidal rule.

### 2.5. Definitions of Treatment Response and Main Variables

This study evaluated treatment response by time to culture conversion, two-month and six-month sputum culture conversion, as well as final treatment outcome. Sputum samples only accompanied with IDs were sent to the respective up-level prefectural TB reference laboratory during the follow-up for laboratory examination. Sputum culture conversion was defined as the occurrence of two consecutive negative cultures with an interval over one month, with the collection date of the first negative culture considered to be the conversion date [28]. The treatment outcome was classified based on the guidelines provided by the WHO [29]. A successful outcome was defined as cure, while a failure outcome was defined as treatment completed without cure, failure, death, and lost to follow-up.

Extensive pulmonary disease was defined by a Timika score (used to assess chest radiograph severity) ≥ 71 [30]. A TB score ≥ 8 [31] was considered to be severe disease. Effective drugs were defined as drugs with DST-confirmed susceptibility and those to which participants had no previous exposure.

### 2.6. Statistical Analyses

The statistics were performed using the R program (version 4.0.0) and IBM SPSS 20.0 (IBM Corp., Armonk, NY, USA). We considered a *p*-value < 0.05 as statistical significance and calculated 95% confidence intervals (CI). To evaluate between-group differences, appropriate methods, such as the Mann–Whitney U-test and χ2 test, were used depending on the dataset.

We compared the distribution of pyrazinamide MIC, AUC_0–24h_, and AUC_0–24h_/MIC between participants with different treatment responses to detect their associations. A multivariate modified Poisson regression model was applied to further evaluate these effects. The correlation between AUC_0–24h_, AUC_0–24h_/MIC and time to sputum culture conversion was evaluated by multivariate Cox proportional hazard regression models and Kaplan–Meier survival analysis.

Univariable regression models were used to clarify risk factors to be adjusted. An ordered logistic regression model was applied for MIC and linear regression was applied for AUC_0–24h_ and AUC_0–24h_/MIC. A modified Poisson regression model was used to calculate the risk ratios for treatment response. The Cox proportional hazard regression model was applied to calculate hazard ratios for culture conversion time. Potential confounders included in the adjusted analysis consisted of area, sex, BMI, smoking, TTP at baseline, severe disease, and Group A drug-based regimens according to the univariable analysis above and the clinical experience. Subgroup analysis was performed among participants receiving different Group A drug-based regimens.

We performed Classification Regression Tree (CART) analysis to identify the key thresholds of pyrazinamide AUC_0–24h_/MIC predictive of treatment response at different moments. It was conducted using Salford Predictive Miner 8.3.2.0 (Salford Systems, San Diego, CA, USA) and presented as intuitive decision trees with the root node as the primary predictor. The percentage of target attainment was calculated. The association between pyrazinamide thresholds and treatment response was analyzed using Kaplan–Meier survival analysis, multivariate modified Poisson regression, as well as Cox proportional hazards regression models. To make the regression analysis statistically feasible, the thresholds were increased so that all groups contained at least one subject. The receiver operating characteristic (ROC) curves were applied to evaluate the performance of the thresholds.

## 3. Results

### 3.1. Study Population

During the study period, there were 374 patients newly diagnosed with MDR-TB, among whom 143 patients were included. Nineteen percent (71/374) of patients were excluded because of having pyrazinamide-resistant isolates detected (Figure 1). Of the 143 participants, the mean ±SD of age was 43.3 ±10.4 years, males accounted for 58.7%, and the median (interquartile range (IQR)) of weight was 58 (50–69) kg (Table 1).

Among one hundred and forty-three participants using pyrazinamide, the regimens used were as follows: thirty-five participants (24.5%) received “moxifloxacin + bedaquiline + linezolid”-based regimens, sixty-four participants (44.8%) received “moxifloxacin + linezolid”-based regimens, thirty-seven participants (25.9%) received “moxifloxacin”-based regimens, and seven participants (4.9%) received “linezolid”-based regimens.

### 3.2. Pyrazinamide Exposure and Susceptibility

More than half (74, 51.1%) of the *M. tuberculosis* isolates from the 143 participants studied had baseline pyrazinamide MIC values of 64 mg/L, which was close to the critical concentration of 100 mg/L [25]. The median (IQR) values for AUC_0–24h_ and the AUC_0–24h_/MIC ratio were 336.8 (230.0–456.1) mg·h/L and 9.7 (4.8–13.7), respectively (Figure 2).

Among one hundred and forty-three participants using pyrazinamide, the regimens used were as follows: thirty-five participants (24.5%) received “moxifloxacin + bedaquiline + linezolid”-based regimens, sixty-four participants (44.8%) received “moxifloxacin + linezolid”-based regimens, thirty-seven participants (25.9%) received “moxifloxacin”-based regimens, and seven participants (4.9%) received “linezolid”-based regimens.

### 3.3. Treatment Responses

During the study, no participants were dead or lost to follow-up. There was a total of 132 adverse events reported, including gastrointestinal disorders (26.6%), peripheral neuropathy (14.7%), psychiatric disorders (12.6%), and OTcF prolongation (11.2%) (Appendix A). Thirteen participants were reported to experience pyrazinamide-induced serious adverse events, such as hepatotoxicity (5, 3.5% of all participants), gastrointestinal intolerance (4, 2.8%), cutaneous adverse reactions (2, 1.4%), and arthropathy (2, 1.4%). The adverse effects showed no significant differences compared to those reported in the previous study [9]. Regimen modification was not conducted because of controlled adverse effects.

The median (IQR) time to culture conversion was 6 (2–24) months. After two months of treatment, sputum culture conversion was achieved in 52 participants (36.4%). The number increased to 79 participants (55.2%) after six months of treatment. Ultimately, 96 participants (67.1%) had a successful treatment outcome after 24 months of treatment (Table 1).

The treatment responses varied among participants taking different Group A drugs (Figure 3). Patients who received all three Group A drugs had a larger percentage of sputum culture conversion at two (65.7%) and six months (82.9%), as well as a successful treatment outcome (85.7%). Conversely, participants who received only one Group A drug (e.g., moxifloxacin) were less likely to achieve sputum culture conversion at two (24.3%) and six months (32.4%), as well as a successful treatment outcome (43.2%).

The univariable analysis revealed that smoking, severe disease, and the use of less than three Group A drugs were risk factors, while longer TTP and residing in Jiangsu were protective factors for all of the evaluated treatment responses (Appendix A).

### 3.4. The Correlation between Exposure and Susceptibility of Pyrazinamide and Treatment Responses

Significant differences were observed in pyrazinamide MIC values of *M. tb* isolates between participants with different sputum culture results (Figure 4). Participants with isolates of lower MIC values were more likely to achieve 2-month sputum culture conversion (MIC = 16 mg/L: 100.0%; MIC = 32 mg/L: 53.8%; MIC = 64 mg/L: 9.5%; *p* < 0.001). The same association also appeared for 6-month sputum culture conversion (MIC = 16 mg/L: 100.0%; MIC = 32 mg/L: 61.5%; MIC = 64 mg/L: 40.5%; *p* < 0.001) and treatment outcome (MIC = 16 mg/L: 100.0%; MIC = 32 mg/L: 80.8%; MIC = 64 mg/L: 50.0%; *p* < 0.001).

Pyrazinamide AUC_0–24h_ varied significantly between participants with positive and negative sputum culture results after two months’ treatment (282.6 vs. 432.7; *p* < 0.001), six months’ treatment (237.4 vs. 446.3; *p* < 0.001), and 24 months’ treatment (245.8 vs. 435.8; *p* < 0.001), with higher drug exposure for those with sputum culture conversion and successful outcome. The median AUC_0–24h_/MIC varied significantly between participants with positive and negative sputum culture results after two months’ treatment (5.2 vs. 13.9; *p* < 0.001), six months’ treatment (4.5 vs. 13.4; *p* < 0.001), and 24 months’ treatment (4.1 vs. 13.0; *p* < 0.001), with larger values in the sputum culture conversion or successful outcome. There was no significant association between AUC_0–24h_ levels and the two-month sputum culture conversion (*p* > 0.05) among participants using all three Group A drugs. However, in all subgroups, a significant association was observed between AUC_0–24h_/MIC levels and two-month and six-month sputum culture conversion, and treatment outcome (*p* < 0.001) (Figure 4). Survival analysis shows the participants with *M. tuberculosis* isolates showing lower pyrazinamide MIC values were more likely to have a shorter time to sputum culture conversion, similar to participants with high AUC_0–24h_ and AUC_0–24h_/MIC (*p* < 0.001) (Figure 5).

After adjusting for geographic area, sex, BMI, smoking, severe disease, time to positivity at baseline, and Group A drug-based regimens, participants with higher pyrazinamide AUC_0–24h_/MIC values had a larger probability of two-month culture conversion (aRR 1.1, 95% CI 1.07–1.20), six-month culture conversion (aRR 1.1, 95% CI 1.06–1.16), treatment success (aRR 1.07, 95% CI 1.03–1.10), and earlier culture conversion time (aHR 1.18, 95% CI 1.14–1.23) (Table 2). In subgroups of Group A drug-based regimens, multivariate analysis showed similar associations between AUC_0–24h_/MIC levels and all evaluated treatment responses. However, pyrazinamide AUC_0–24h_/MIC had greater RR or HR values in participants who used fewer Group A drugs compared with those who used three Group A drugs.

### 3.5. CART Analysis of Pyrazinamide Exposure/Susceptibility Target

CART analyses were performed among all participants (Figure 6). The primary node for two-month culture conversion was pyrazinamide AUC_0–24h_/MIC of 12.7 (39.2% achieved), where 92.9% of participants achieving this target had a two-month culture conversion, compared with none in those below the target. Similarly, the primary nodes for the six-month culture conversion and treatment outcome were pyrazinamide AUC_0–24h_/MIC of 7.8 (55.2% achieved) and 6.3 (58.7% achieved), respectively.

Patients with 7.8 ≤ AUC_0–24h_/MIC < 12.7 all achieved sputum culture conversion within six months with one or two Group A drugs used. If three group A drugs were used, the sputum culture of participants with AUC_0–24h_/MIC in a range of 6.3–7.8 completely achieved sputum culture conversion within eight months, and that with AUC_0–24h_/MIC 7.8–12.7 was achieved within four months. An AUC_0–24h_/MIC higher than 12.7 shortened the time to sputum culture conversion by two months. An AUC_0–24h_/MIC less than 6.3 predicted a less successful treatment outcome (Figure 7).

After modifying thresholds and adjusting for geographic area, sex, BMI, smoking, severe disease, time to positivity at baseline, and Group A drug-based regimens (Table 3), participants with pyrazinamide exposure above those CART-derived thresholds had a greater probability of two-month (1.1% vs. 92.7%, aRR: 77.8, 95% CI: 10.7–546.3) and six-month culture conversion (1.5% vs. 100.0%, aRR:67.8, 95% CI: 10.5–436.5), successful treatment outcome (22.0% vs. 98.8%, aRR: 4.2, 95% CI: 2.6–6.9), and achieving earlier culture conversion (Figure 7).

### 3.6. CART-Derived Threshold Performance Evaluation

The area under the ROC curves (AUC score) of two- and six-month culture, as well as treatment outcome, was 0.978, 1, and 0.922, respectively (Table 4, Figure 8), indicating that the CART-derived thresholds were good predictors of treatment responses. AUC_0–24h_/MIC = 7.8 showed great predictive performance for six-month culture conversion (AUC = 1) no matter what Group A drugs were used. All three CART-derived thresholds were predominant predictors of respective treatment responses when treated with three Group A drugs (AUC = 1).

## 4. Discussion

This study demonstrated an increased probability of sputum culture conversion and a successful treatment outcome with higher ratios of pyrazinamide AUC_0–24h_ divided by the MIC for *M. tuberculosis* during MDR-TB treatment in programmatic regimens. This study also identified clinical thresholds of pyrazinamide at three time points: 12.7 (two months), 7.8 (six months), and 6.3 (treatment outcome at 24 months). Furthermore, clinical recommendations for pyrazinamide were made based on the test of AUC_0–24h_ and MIC.

A similar range of pyrazinamide AUC_0–24h_ was observed and the variations may be caused by the heterogeneity of the population, dosage, comorbidity, and other related factors [6,10,12]. This study observed significant differences in AUC_0–24h_ levels among patients with different treatment responses, and higher exposure levels were associated with shorter times to sputum culture conversion. These findings are consistent with previous studies [10,13]. However, in this study, the aforementioned correlation was not significant among patients who were treated with Group A drugs extensively. Therefore, AUC_0–24h_ may not be a perfect clinical target for pyrazinamide.

During the study period, *M. tuberculosis* isolates in 19.0% (71/374) of participants were resistant to pyrazinamide, which is lower than 40% in the previous study conducted in Chinese patients with MDR-TB [9]. This may be attributed to the fact that more than 80% of the registered patients during this study were newly diagnosed and no previous usage of pyrazinamide may lead to a relatively lower resistance to pyrazinamide [18]. On the other hand, this study shows that the majority of patients with MDR-TB and *M. tuberculosis* isolates resistant to pyrazinamide have high-level MICs. In this study, lower MIC levels of pyrazinamide were significantly associated with better treatment responses. This finding is inconsistent with some previous studies [8,14]. This discrepancy may be explained by the fact that our study was conducted in a larger population of patients with phenotypically pyrazinamide-sensitive MDR-TB. We also excluded some patients to avoid the impact of confounders. We analyzed MICs as an ordered categorical variable three levels below the critical concentration. Therefore, the present study suggested that MIC testing proves valuable in guiding the treatment, even among patients with isolates phenotypically sensitive to pyrazinamide.

This study further suggested that a higher AUC_0–24h_/MIC was associated with better efficacy, consistent with previous research [6,7,8,9,22]. This association remained significant regardless of the number of Group A drugs utilized and was more pronounced in patients treated with fewer than three Group A drugs. Therefore, AUC_0–24h_/MIC is considered a better clinical target parameter for pyrazinamide, taking into account its general applicability across patients using various regimens. Monitoring AUC_0–24h_/MIC may be given greater emphasis, particularly in patients using fewer Group A drugs.

This study identified three specific AUC_0–24h_/MIC values (12.7, 7.8, 6.3). Among these thresholds, AUC_0–24h_/MIC = 12.7 (achieving a rate of 39.2%) is a good indicator for predicting sputum culture conversion at two months. However, it can only be achieved in a small number of patients with lower MIC levels or good pyrazinamide absorption, so it is not strongly recommended. AUC_0–24h_/MIC = 6.3 (achieving a rate of 58.7%) is associated with treatment outcomes at 24 months, but its predictive performance is moderate. AUC_0–24h_/MIC = 7.8 (achieving a rate of 55.2%) is a good indicator for predicting sputum culture conversion at six months, balancing feasibility and shorter treatment duration. This threshold is slightly lower than the previously established recommendation in DS-TB, but comes with a significantly prolonged treatment duration (AUC_0–24h_/MIC > 8.42 as TDM target for 2 weeks of culture conversion [22]). However, only 55.2% of participants in the study population were able to reach the threshold using the dosage of pyrazinamide recommended by the WHO, possibly due to high MIC levels. The pyrazinamide AUC_0–24h_/MIC threshold may be selected according to Group A drug-based regimens. In the case of insufficient Group A drugs, a higher threshold is required to achieve the same effect of shortening the culture conversion time. On the other hand, the phenomenon also provides room for patients experiencing pyrazinamide-related adverse events due to high drug exposure to reduce the dosage of pyrazinamide if more Group A drugs are used.

Considering the association between the AUC_0–24h_/MIC of pyrazinamide and treatment responses, we propose recommendations to optimize the use of pyrazinamide in MDR-TB treatment. First of all, quantitatively testing the level of resistance (i.e., MIC) to pyrazinamide, instead of only the DST, may be conducted. Switching to other effective drugs may be of high priority in patients with isolates borderline resistant to pyrazinamide. Furthermore, for patients diagnosed with high-MIC isolates but still using pyrazinamide, dosage may be individually adjusted by combining the blood concentration with the MIC to trade off between efficacy and toxicity. The pyrazinamide exposure/susceptibility threshold (i.e., AUC_0–24h_/MIC = 7.8 in the present study) should undergo further evaluation in subsequent clinical trials to value its effectiveness in guiding treatment decisions. We believe that lower pyrazinamide AUC_0–24h_/MIC levels contribute to treatment failure, which has also been observed for other anti-tuberculosis drugs, especially Group A drugs. Therefore, if treatment failure occurs, it is necessary to reassess the level of drug susceptibility of *M. tb* strains, transition to a regimen comprising anti-TB drugs to which the strains are sensitive, and optimize the dose for sufficient drug exposure. It is also necessary to assess treatment adherence, comorbidities, and nutritional status.

One of the notable strengths of this study is the validation of the impact of pyrazinamide exposure/susceptibility on treatment response in a real multicenter population. This study further indicated that the appropriate use of pyrazinamide can lead to enhanced efficacy and subsequently better treatment outcomes than the general long-term regimen, consistent with previous controlled clinical studies in patients with MDR-TB [7]. In this study, the CART-derived AUC_0–24h_/MIC thresholds of pyrazinamide were constructed based on real pharmacokinetic data. By utilizing these thresholds, patients can potentially adjust their regimens to achieve adequate levels of drug exposure, thereby enhancing treatment efficacy. This study also analyzed the effect of group A drugs on the efficacy of pyrazinamide in patients with MDR-TB, providing recommendations for personalized TDM based on drug combinations.

There are some limitations in this study. Firstly, we included specific patients with MDR-TB to control the impact of the possible confounders. Therefore, the study conclusions can be directly applicable to the majority of adult MDR-TB patients in good condition. The findings may also be generalized to populations with specific conditions (such as hepatitis, HIV, diabetes, and hepatic or renal impairment) by adjusting for potential confounding factors related to both drug exposure and treatment responses. Secondly, the predictive performance of the thresholds was not analyzed in larger validation cohorts. Utilizing limited sampling strategies and population pharmacokinetics models is a more effective approach than intensive blood sampling in small clinical sample sizes [11] for establishing extensive development and validation cohorts. Finally, this study only explored the target values of pyrazinamide in the context of MDR-TB due to the limited data, and the same approach can be extended to the study of other promising drugs.

## 5. Conclusions

Our study suggests that the drug exposure/susceptibility ratio of pyrazinamide is significantly correlated with treatment response in patients with MDR-TB. A pyrazinamide AUC_0–24h_/MIC ≥ 7.8 was identified as the best predictor for discerning the six-month sputum culture conversion. In the treatment of MDR-TB, drug exposure and quantitative susceptibility testing are still required to determine the use of pyrazinamide and ensure the attainment of clinical thresholds, especially for patients with *M. tb* isolates showing borderline resistance to pyrazinamide. Randomized controlled studies and further studies in short-course regimens are recommended to validate the clinical targets of pyrazinamide.

## Figures and Tables

**Figure 1 pharmaceutics-16-00144-f001:**
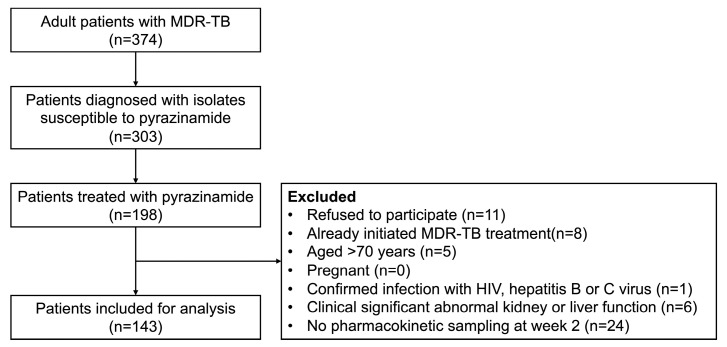
The enrolment process of participants with multidrug-resistant tuberculosis. MDR-TB: Multidrug-Resistant Tuberculosis; HIV: Human Immunodeficiency Virus.

**Figure 2 pharmaceutics-16-00144-f002:**
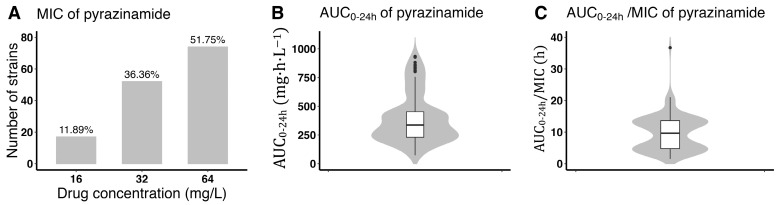
The distribution of MIC (**A**), AUC_0–24h_ (**B**), and AUC_0–24h_/MIC ratio (**C**) of pyrazinamide in participants with multidrug-resistant tuberculosis.

**Figure 3 pharmaceutics-16-00144-f003:**
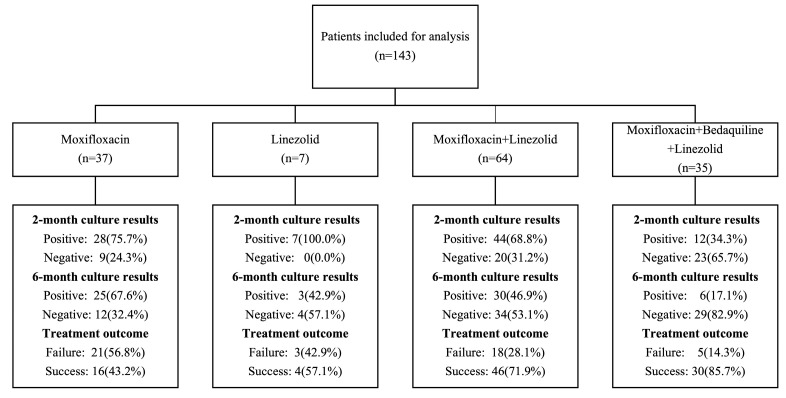
Treatment responses in participants with multidrug-resistant tuberculosis receiving different Group A drug-based regimes.

**Figure 4 pharmaceutics-16-00144-f004:**
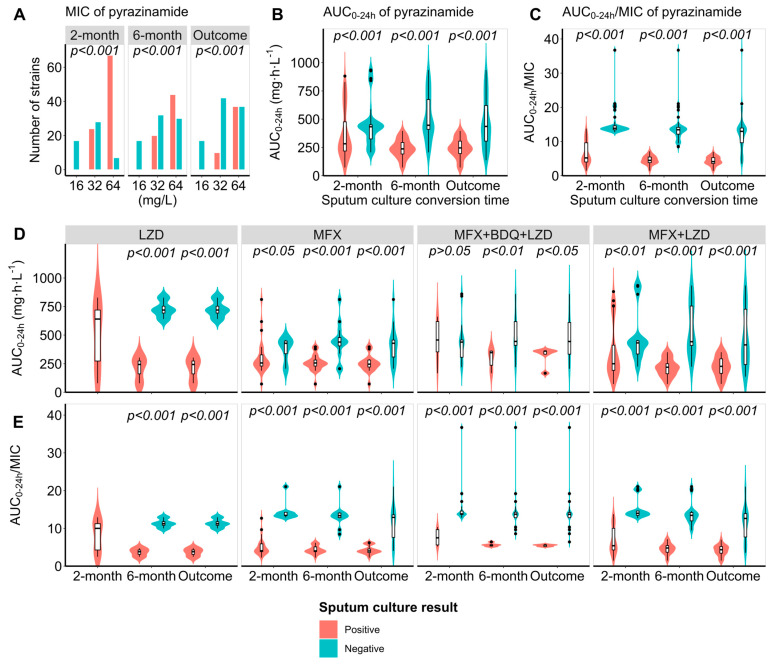
The distribution of MIC (**A**), AUC_0–24h,_ (**B**,**D**), and AUC_0–24h_/MIC ratio (**C**,**E**) of pyrazinamide with different treatment responses in patients with multidrug-resistant tuberculosis. MFX: moxifloxacin; LZD: linezolid; BDQ: bedaquiline.

**Figure 5 pharmaceutics-16-00144-f005:**
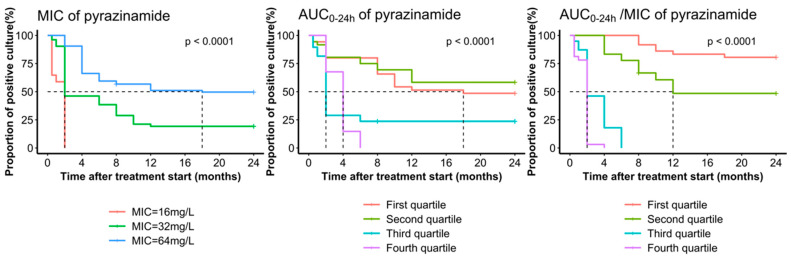
Time to culture conversion among patients with multidrug-resistant tuberculosis grouped by MIC levels, AUC_0–24h_, and AUC_0–24h_/MIC ratio quartiles of pyrazinamide. First quartile: 25% of smallest numbers; Second quartile: between 25.1% and 50%; Third quartile: 50.1% to 75%; Fourth quartile: 25% of the largest numbers. Dotted line: median survival time at which 50% of the participants had still not achieved sputum culture conversion.

**Figure 6 pharmaceutics-16-00144-f006:**
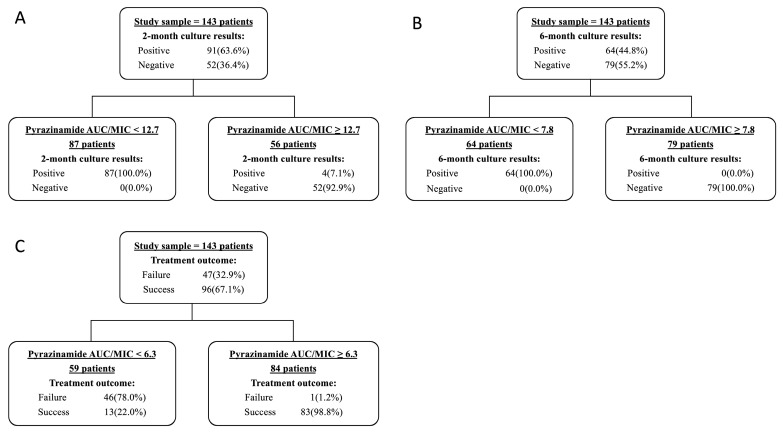
Random forest and Classification and Regression Tree (CART) analysis for two-month sputum culture results (**A**), six-month sputum culture results (**B**), and treatment outcome (**C**).

**Figure 7 pharmaceutics-16-00144-f007:**
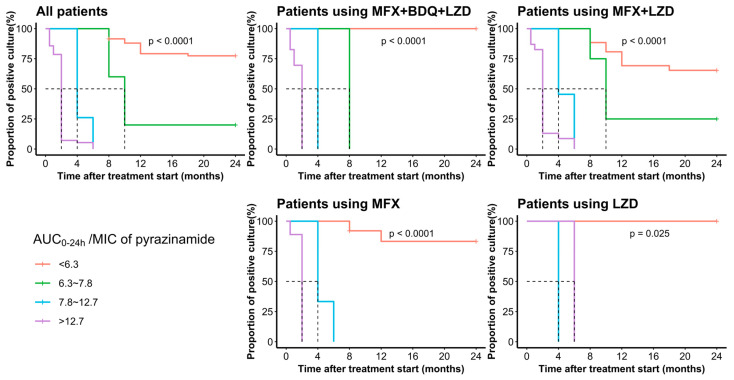
Time to culture conversion in patients with multidrug-resistant tuberculosis grouped by CART-derived thresholds of pyrazinamide AUC_0–24h_/MIC. MFX: moxifloxacin; LZD: linezolid; BDQ: bedaquiline. Dotted line: median survival time at which 50% of participants did not achieve sputum culture conversion.

**Figure 8 pharmaceutics-16-00144-f008:**
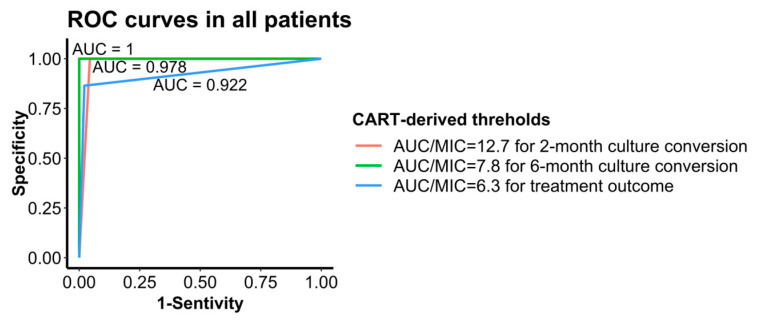
ROC curves for different thresholds to treatment responses with all patients with MDR-TB. ROC: receiver operating characteristic; AUC: the area under the receiver operating characteristic (ROC) curves.

**Table 1 pharmaceutics-16-00144-t001:** Demographic characteristics, clinical features, treatment regimens, and responses among the participants with multidrug-resistant tuberculosis using pyrazinamide (*n* = 143).

Characteristics	Number (%)
Area	
Guangzhou, China	43 (30.1)
Henan, China	50 (35.0)
Jiangsu, China	50 (35.0)
Age (years)	43.3 ± 10.4
Male	84 (58.7)
Weight (kg)	58.0 (50.0–69.0)
BMI (kg/m^2^)	20.9 ± 2.7
Current smoker	82 (57.3)
Diabetes mellitus type II	27 (18.9)
Pulmonary cavities	54 (37.8)
Extensive pulmonary disease (Timika score ≥ 71)	0 (0.0)
Severe disease (TB score ≥ 8)	31 (21.7)
Culture time to positivity (days)	11.9 ± 3.3
Drug intake	
moxifloxacin (400 mg, once daily)	136 (95.1)
linezolid (600 mg, once daily)	106 (74.1)
bedaquiline (400 mg, once daily)	35 (24.5)
Clofazimine (100 mg, once daily) ^a^	109 (76.2)
Cycloserine (500 mg, once daily)	129 (90.1)
P-aminosalicylic acid (3.3–6.6 g, twice a day)	9 (6.3)
Prothionamide (600 mg, three times a day)	41 (28.7)
pyrazinamide (1500 mg, three times a day)	143 (100.0)
Ethambutol (750 mg, once daily)	7 (4.9)
Pyrazinamide dosage (mg/kg)	26.9 ± 5.7
Using five effective drugs ^b^	143 (100.0)
Group A drugs ^c^	
moxifloxacin + bedaquiline + linezolid	35 (24.5)
moxifloxacin + linezolid	64 (44.8)
moxifloxacin	37 (25.9)
linezolid	7 (4.9)
Two-month culture conversion	52 (36.4)
Six-month culture conversion	79 (55.2)
Time to culture conversion (months)	6 (2–24)
Treatment outcome ^d^	
Success	96 (67.1)
Failure	47 (32.9)

Data are present as mean ± SD, median (IQR), or No. (%); BMI: Body Mass Index; ^a^: loading dosage 200 mg twice daily for two months; ^b^: Effective drugs referred to drugs with confirmed susceptibility by phenotypic DST or no previous exposure history; ^c^: Group A drugs were identified according to MDR-TB treatment guidelines recommended by WHO [4]; ^d^: a successful outcome was defined as cure, while a failure outcome was defined as treatment completed without cure, failure, death, and lost to follow-up [28].

**Table 2 pharmaceutics-16-00144-t002:** Multivariate analysis for AUC_0–24h_/MIC in patients with multidrug-resistant tuberculosis and different Group A drug-based regimens.

Treatment Regimen	Two-Month Culture Conversion		Six-Month Culture Conversion		Treatment Outcome		Time to Culture Conversion	
	Negative No. (%) ^a^	Adjusted RR ^b^ (95% CI)	Negative No. (%) ^a^	Adjusted RR ^b^ (95% CI)	Success No. (%) ^a^	Adjusted RR ^b^ (95% CI)	Month Median (IQR)	Adjusted HR ^b^ (95% CI)
All patients	52 (36.4)	1.1 (1.09–1.20)	79 (55.2)	1.1 (1.06–1.16)	96 (67.1)	1.07 (1.03–1.10)	6 (2–24)	1.18 (1.14–1.23)
MFX + LZD + BDQ	23 (65.7)	1.1 (1.03–1.12)	29 (82.9)	1.05 (1.02–1.08)	30 (85.7)	1.04 (1.01–1.07)	2 (2–4)	1.12 (1.05–1.21)
MFX + LZD	20 (31.3)	1.5 (1.31–1.62)	34 (53.1)	1.2 (1.14–1.26)	46 (71.9)	1.1 (1.05–1.12)	6 (2–24)	1.6 (1.44–1.86)
MFX	9 (24.3)	3.0 (2.03–4.22)	12 (32.4)	1.4 (1.26–1.65)	16 (43.2)	1.2 (1.06–1.26)	24 (3–24)	2.2 (1.44–3.40)
LZD	0 (0.0)	1.0 (0.79–1.25) ^c^	4 (57.1)	21.0 (17.84–24.63)	4 (57.1)	21.0 (17.84–24.63)	6 (4–24)	153 (23.57–1036)

^a^: the proportion of patients with sputum culture conversion in patients using specific Group A drug-based regimen; ^b^: adjusted according to the area, sex, BMI, smoking, time to positivity, severe disease, and Group A drug-based regimens; ^c^: *p* > 0.05; MFX: moxifloxacin; LZD: linezolid; BDQ: bedaquiline; RR: risk ratio; HR: hazard ratio; CI: confidence interval; IQR: interquartile range.

**Table 3 pharmaceutics-16-00144-t003:** Univariate and multivariate analysis for AUC_0–24h_/MIC grouped by CART-derived threshold.

CART-Derived Threshold	Sputum Culture Conversion/Treatment Outcome			Time to Culture Conversion		
	Success/negative No. (%) ^a^	RR (95% CI)	Adjusted RR ^b^ (95% CI)	Month Median (IQR)	HR (95% CI)	Adjusted HR ^b^ (95% CI)
Two-month						
AUC_0–24h_/MIC ≤ 12.8 ^c^	1 (1.1)	1	1	24 (6–24)	1	1
AUC_0–24h_/MIC > 12.8 ^c^	51 (92.7)	81.6 (11.6–573.6)	77.8 (10.7–564.3)	2 (2–2)	20.4 (11.8–35.3)	24.7 (12.9–47.3)
Six-month						
AUC_0–24h_/MIC ≤ 8.5 ^d^	1 (1.5)	1	1	24 (12–24)	1	1
AUC_0–24h_/MIC > 8.5 ^d^	78 (100.0)	65.0 (9.3–454.5)	67.8 (10.5–436.5)	2 (2–4)	302.5 (40.2–2273.1)	334.2 (43.4–2576.3)
Treatment outcome						
AUC_0–24h_/MIC ≤ 6.3	13 (22.0)	1	1	24 (18–24)	1	1
AUC_0–24h_/MIC > 6.3	83 (98.8)	4.5 (2.8–7.3)	4.2 (2.6–6.9)	2 (2–4)	28.2 (14.5–55.2)	30.6 (14.5–64.8)

^a^: the proportion of patients with sputum culture conversion in patients using specific Group A drug-based regimen; ^b^: adjusted according to the area, sex, BMI, smoking, time to positivity, severe disease, and Group A drug-based regimens; ^c^: the threshold of 12.7 was increased so that all groups contained at least one subject for statistical feasibility; ^d^: the threshold of 7.8 was increased so that all groups contained at least one subject for statistical feasibility; RR: risk ratio; HR: hazard ratio; CI: confidence interval; IQR: interquartile range.

**Table 4 pharmaceutics-16-00144-t004:** The areas under the ROC curves for CART-derived AUC_0–24h_/MIC thresholds.

Treatment Regimen	AUC_0–24h_/MIC = 6.3 (for Treatment Outcome)	AUC_0–24h_/MIC = 7.8 (for Six-Month Culture Conversion)	AUC_0–24h_/MIC = 12.7 (for Two-Month Culture Conversion)
MFX + LZD + BDQ	1.000	1.000	1.000
MFX + LZD	0.874	1.000	0.966
MFX	0.875	1.000	1.000
LZD	/	1.000	1.000

MFX: moxifloxacin; LZD: linezolid; BDQ: bedaquiline.

## Data Availability

The data can be shared up on request.

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
