# Peer review of "Drug Exposure and Susceptibility of Pyrazinamide Correlate with Treatment Response in Pyrazinamide-Susceptible Patients with Multidrug-Resistant Tuberculosis"

_pharmaceutics, 2024, doi:10.3390/pharmaceutics16010144_

Round 1
Reviewer 1 Report
Comments and Suggestions for Authors
The study entitled ‘Drug Exposure and Susceptibility of Pyrazinamide Correlate with Treatment Response in Pyrazinamide-Susceptible Patients with Multidrug-Resistant Tuberculosis’ links pyrazinamide drug exposure/susceptibility ratios to treatment outcomes in multidrug-resistant tuberculosis (MDR-TB) patients. It identifies a crucial pyrazinamide AUC0-24h/MIC threshold of 7.8, impacting sputum culture results after two months. However, practical implications remain unclear. The study underscores the importance of drug testing, especially in susceptible M. tuberculosis isolates. Establishing clinical targets based on Group A drug usage is suggested. Further research is recommended, though specific directions are lacking. However, before accepting this work, several issues must be addressed, with particular emphasis on the points listed below.
1. Introduction
a. While the text mentions China's TB burden and issues with pyrazinamide resistance, it could benefit from a broader global perspective on the MDR-TB problem and the use of pyrazinamide in different regions.
b. The text repeats information multiple times, such as the importance of pyrazinamide, drug exposure, and resistance levels. This repetition makes the text unnecessarily long and repetitive.
c. The text could be seen as overly negative, focusing on the challenges and problems related to MDR-TB treatment in China without offering potential solutions or highlighting any positive developments.
2. Materials and methods
a. The inclusion and exclusion criteria are well-defined and appear appropriate for the study. However, it would be helpful to provide a brief rationale for each criterion to justify its inclusion or exclusion. Specify whether there was any randomization or blinding involved in participant selection or treatment allocation. If not, justify the lack of randomization.
b. Please specify the number of participants included in the study. This information is crucial for understanding the study's sample size and provide more details on the selection criteria for the three designated hospitals in Guangzhou, Henan, and Jiangsu Province. Were these hospitals chosen randomly, or were there specific reasons for their selection?
c. Clarify if the response to treatment outcomes (e.g., two-month sputum culture conversion) were assessed by an independent blinded assessor or if there was potential for bias in these assessments.
d. Explain why the analytical range of pyrazinamide was set at 0.1 to 100 mg/L and provide references, if applicable.
e. Explain the relevance and importance of the pharmacokinetic parameters of pyrazinamide, especially AUC0-24h, in the context of MDR-TB treatment. Mention whether there were any adjustments or modifications made to the treatment based on the pharmacokinetic results.
3. Result
a. While the discussion of pyrazinamide susceptibility is informative, it lacks a broader context. Are these susceptibility rates consistent with previous studies? How do they compare to the general population of MDR-TB patients? Without such comparisons, it's challenging to assess the significance of these findings.
b. he reporting of adverse events is appreciated, but the paper does not discuss whether these adverse events were expected based on previous research. Were these rates of adverse events higher or lower than in similar studies?
c. The sudden switch to other group C drugs for participants experiencing pyrazinamide-induced serious adverse events raises questions about the robustness of the treatment regimen. What were the implications of this switch for the study outcomes, and were there any long-term consequences?
d. In the section “The Association between pyrazinamide Exposure, Susceptibility, and Treatment Responses”: The variation in MIC values between different treatment outcomes is presented, but the discussion lacks depth. It's unclear why lower MIC values are associated with negative culture results, and this warrants further exploration.
e. The significance of MIC values in TB treatment could be better explained. Additionally, it's unclear whether these findings have practical implications for treatment strategies.
4. Discussion
a. The discussion lacks context and fails to relate the findings to the broader field of MDR-TB treatment. It's essential to discuss how these results contribute to the existing body of knowledge and whether they challenge or reinforce current treatment guidelines.
b. The study acknowledges the need for further validation in other populations, but it doesn't delve into the specifics of how these findings might or might not apply to different patient demographics or regions. This lack of consideration limits the practical utility of the research.
c. The study mentions clinical recommendations without providing specific guidance or actionable advice for healthcare practitioners. It's essential to offer practical insights into how these findings can be applied in a clinical setting.
d. The conclusion mentions that the use of pyrazinamide in patients with susceptible isolates requires drug exposure and susceptibility testing, but it doesn't provide specific guidance on how these tests should be conducted or how results should inform treatment decisions.
Author Response
Dear editors and reviewers of Pharmaceutics
Thank you for your letter and for the reviewers’ comments concerning our manuscript entitled “Drug Exposure and Susceptibility of Pyrazinamide Correlate with Treatment Response in Pyrazinamide-Susceptible Patients with Multidrug-Resistant Tuberculosis”. These comments help revise and improve our manuscript.
We studied the comments carefully and made revisions in correspondence to the comments. All changes in the paper have been highlighted and we provided our point-by-point responses to the reviewers’ comments in “Response to reviewers”.
Thank you and best regards
Jia Liu
The Fifth People’s Hospital of Suzhou
Infectious Disease Hospital Affiliated to Soochow University
No.10 Guangqian Road,
Suzhou, China
Email: [email protected]

Reviewer 2 Report
Comments and Suggestions for Authors
I ma happy with the paper in its present format. I feel that it makes a significant contribution to the field. I am not a statistician so I am not able to determint if there are more appropriate ways of analysing the data. The methods used appear to be quite sound and the conclusions are a logical step. This paper does not claim to be the definitive answer to the problem but it makes a very valuable contribution.
One question I do have is what the authors think the appropriate course of action is for those patients that fail to respond to the antibiotic therapies and are still infected with XDR organisms at the conclusion of the routine therapy. This was not addressed. Some speculation on the most appropriate course of action would be interesting.
Comments on the Quality of English LanguageLine 26 Finding clinical thresholds
Line 75 derive a pyrazinamide threshold
Line 210 should that read (50 -69)?
Line 422 determine the MIC at baseline?
Author Response

(The authors gave the same response as above.)

Reviewer 3 Report
Comments and Suggestions for Authors
Page 6/17- under table no. 1(LINE 223-227) you have mentioned about different regimens including DRTB drugs as in moxifloxacin + linezolid + bedaquilline, moxifloxacin + linezolid , moxifloxacin based regimen
However no guidelines recommend any such regimen because designing such a regimen will be grossly inadequate for the treatment of DRTB patients
Furthermore availability of pattern of resistance is highly desirable in form of first line and second line LPA to design an optimized regimen for treatment of such patients ; but you have only considered the sputum culture in your study
Comments on the Quality of English LanguageIn bracket contains the –to be replaced content .
Page 2/17
Line 44- it is a threat to the (threatening)
Line 50- it is important to optimize the use of currently available drugs.(therefore optimizing)
Line 53- decrease in relapse rates(decreased relapse)
Line 55- reducing time of sputum culture(time to sputum)
Line 56- increase in successful treatment outcomes(increase successful treatment)
Line 57- explained in terms of drug exposure ; given the standard doses, the drug exposure of pyrazinamide (explained by drug)
Line 58- given the standard doses(given standard doses)
Line 70- threshold in TDM in TB treatment(TDM of TB treatment)
Line 72- derived its threshold to be(derived threshold 8.42)
Line 73- was then cited as a pyrazinamide (was cited)
Line 77- The threshold of pyrazinamide may require re-evaluation with respect to the currently recommended treatment (pyrazinamide threshold, with the currently recommended treatment)
Line 81- In china because of the problems such as(because of problems)
Line 85- to fill this knowledge gap(the)
Page 3/17
Line 105- designated hospitals for two weeks of inpatient treatment(two weeks inpatient)
Line 108- Cycloserine for 18 months(continuation phase). (consolidation)
Line 109- on the basis of (if appropriate according to)
Line 115- as well as other reasons were recorded. Patients were examined (as well as reasons)
Line 116- continuation(consolidation)
Line 119- for regular quality (where regular quality )
Line 122- was defined as the time between collection of the culture and positive signal in BACTEC system. This was considered as a marker of(was defined as time to culture positivity)
Line 128- isolates that were pyrazinamide-susceptible via the phenotypic drug(isolates being diagnosed)
Line 134- Bacterial suspension which was inoculated(suspension was inoculated)
Line 141- venous catheter initially at pre-dose as well (venous catheter at predose)
Page 4/17
Line 166- defined by the TIMIKA score(defined as the)
Line 177- applied for MIC and linear regression was applied(the linear regression)
Line 181- responses were compared(responses was)
Line 189- analysis above and the clinical experience(above and clinical experience)
Page 5/17
Line 207- during the study period amongst which 143 of them (period and 143 of them)
Line 208- 102 patients were excluded which included 71 patients who were diagnosed with (patients excluded including 71 patients)
Line 210- age/ weight were(of weight was)
Right box- 5th point of sequence of exclusion – confirmed infection with HIV(HCV)
Page 7/17
Line 244- during the treatment, 13 study participants(cut “ of the “)
Line 245- CUT “ toxicity present “
Line 246- such as hepatotoxicity
Line 248- adverse events reversed (cut “ recovered”)
Page 13/17
Line 374- by the MIC for M. tuberculosis( in place “of “)
Page 14/17
Line 398- drawn for (in place of “ performed of”)
Line 413- served as (in place of “ served”)
Line 422- cut “detect” ; rather than just the phenotypic (add “the”)
Line 423- cut ‘might” and add “may”
Line 425 – shortening the culture (add place of“the” )
Line 431- can lead to enhanced (In place of “early “)
Line 443- extrapolated in (in place of “ to “)
Author Response

(The authors gave the same response as above.)

Reviewer 4 Report
Comments and Suggestions for Authors
I congratulate the authors on presenting their work in an excellent manner. I thoroughly enjoyed reading and reviewing their manuscript. The study is straightforward, and the authors have made a commendable effort to include all the factors related to the correlation between the drug and susceptibility of Pyrazinamide with treatment response in pyrazinamide-susceptible patients. Please find the attached document for the correction of minor errors.

Author Response

(The authors gave the same response as above.)

Round 2
Reviewer 1 Report
Comments and Suggestions for Authors
The authors have addressed all my queries.